# Effect of a Fiber D-Limonene-Enriched Food Supplement on Intestinal Microbiota and Metabolic Parameters of Mice on a High-Fat Diet

**DOI:** 10.3390/pharmaceutics13111753

**Published:** 2021-10-21

**Authors:** Maria Chiara Valerii, Silvia Turroni, Carla Ferreri, Michela Zaro, Anna Sansone, Alessandro Dalpiaz, Giada Botti, Luca Ferraro, Renato Spigarelli, Irene Bellocchio, Federica D’Amico, Enzo Spisni

**Affiliations:** 1Department of Biological, Geological and Environmental Sciences, Alma Mater Studiorum-University of Bologna, Via Selmi 3, 40126 Bologna, Italy; chiaravalerii@hotmail.it (M.C.V.); michela.zaro@studio.unibo.it (M.Z.); renato.spigarelli@studio.unibo.it (R.S.); irene.bellocchio2@studio.unibo.it (I.B.); 2Department of Pharmacy and Biotechnology, Alma Mater Studiorum-University of Bologna, Via Selmi 3, 40126 Bologna, Italy; silvia.turroni@unibo.it; 3ISOF, Consiglio Nazionale delle Ricerche, Via P. Gobetti 101, 40129 Bologna, Italy; carla.ferreri@isof.cnr.it (C.F.); anna.sansone@isof.cnr.it (A.S.); 4Department of Chemical, Pharmaceutical and Agricultural Sciences, University of Ferrara, Via Fossato di Mortara 19, 44121 Ferrara, Italy; dla@unife.it (A.D.); giada.botti@unife.it (G.B.); 5LTTA Center, Department of Life Sciences and Biotechnology, University of Ferrara, Via L. Borsari 46, 44121 Ferrara, Italy; frl@unife.it; 6Department of Medical and Surgical Sciences, University of Bologna, Via Zamboni 33, 40138 Bologna, Italy; federica.damico8@unibo.it

**Keywords:** D-Limonene, high fat diet (HFD), intestinal microbiota, obesity

## Abstract

Several studies showed that D-Limonene can improve metabolic parameters of obese mice via various mechanisms, including intestinal microbiota modulation. Nevertheless, its effective doses often overcome the acceptable daily intake, rising concerns about toxicity. In this study we administered to C57BL/6 mice for 84 days a food supplement based on D-Limonene, adsorbed on dietary fibers (FLS), not able to reach the bloodstream, to counteract the metabolic effects of a high-fat diet (HFD). Results showed that daily administration of D-Limonene (30 and 60 mg/kg body weight) for 84 days decreased the weight gain of HFD mice. After 84 days we observed a statistically significant difference in weight gain in the group of mice receiving the higher dose of FLS compared to HFD mice (35.24 ± 4.56 g vs. 40.79 ± 3.28 g, *p* < 0.05). Moreover, FLS at both doses tested was capable of lowering triglyceridemia and also fasting glycemia at the higher dose. Some insights on the relevant fatty acid changes in hepatic tissues were obtained, highlighting the increased polyunsaturated fatty acid (PUFA) levels even at the lowest dose. FLS was also able to positively modulate the gut microbiota and prevent HFD-associated liver steatosis in a dose-dependent manner. These results demonstrate that FLS at these doses can be considered non-toxic and could be an effective tool to counteract diet-induced obesity and ameliorate metabolic profile in mice.

## 1. Introduction

Obesity is a chronic relapsing multifactorial disease affecting about 13% of adults worldwide and is one of the leading risk factors for premature death [1]. There are multiple causes that lead to the deregulation of neuroendocrine mechanisms that regulate energy balance and body weight. Among these, genetic, epigenetic and environmental ones are well recognized [2]. Sedentary life and high consumption of sugars and fats are closely associated with this disease. Dysfunction of the adipose organ, systemic inflammation and alteration of the intestinal microbiota contribute to chronicization and the onset of complications related to obesity that arise over time [1]. The management of the disease based solely on patient education and modification of diet and lifestyle, the lack of effective therapeutic strategies and the complexity of the bariatric surgery process make this pathology difficult to reverse. In obese subjects, there is a rise of morbidity for pathologies such as type II diabetes, cardiopathy and ictus [3]. The intestinal microbiota, capable of modulating various aspects of human physiology, including energy homeostasis [4], is recognized to play a major role in obesity development and maintenance [5]. In fact, it has been shown that, in germ-free mice, fecal transplantation from obese donors resulted in weight gain not dependent on food intake [6]. High sugar and fat consumption and a sedentary lifestyle are obviously the basis of body weight gain; however, it is now clear that the obesity-associated dysbiosis is not to be considered a mere consequence of the quantity and type of foods introduced, but can itself contribute to the progression of this pathological condition and its evolution [7]. Thus, modulating the gut microbiota in these patients could be an interesting therapeutic strategy for obesity management [8]. In recent years, scientific research has focused on the study of numerous substances of natural origin apparently capable of promoting weight loss [9]. Among them, there is a rising interest in orange essential oil (EO) and its main component, D-Limonene, which has shown anti-obesity effects also mediated by its modulatory effect on the gut microbiota of rodents fed a high-fat diet (HFD) [10,11,12]. However, the daily oral dosages at which the orange EO and D-Limonene exerted their anti-obesity effects are quite high and often higher than the proposed maximum dose level value of 25 mg/kg for all animal species [13]. In Wistar rats, orally administered D-Limonene is rapidly absorbed in the gastrointestinal tract and subsequently distributed by the blood to various organs, such as the liver, lung and kidney, where it is readily metabolized [14]. Chronic exposure for 45 days to D-Limonene at daily doses in the range of 25–75 mg/kg body weight caused dose-dependent liver parenchymal damage consisting of steatosis, hepatic cell necrosis and fibrosis [15].

For these reasons we have formulated a fiber D-Limonene-enriched food supplement (FLS), in which the presence of fibers strongly decreases D-Limonene intestinal absorption, thus decreasing its systemic toxicity. In this study, we tested FLS in a murine obesity model since we have already observed, by using an in vitro colon model, that this supplement was capable of positively modulate the colon microbiota [16]. Thus, FLS could be a good candidate for a microbiota-based therapy that counteracts obesity-related alterations in both HFD-fed mice and obese humans.

## 2. Materials and Methods

### 2.1. Chemicals

D-Limonene (≥98%) was provided by Xeda International (S. Andiol, France), HPLC-grade methanol, acetonitrile, ethyl acetate and water were acquired from Sigma-Aldrich (Milan, Italy). All other chemicals were of the highest purity and are commercially available. Fiber D-Limonene-enriched food supplement (FLS), made by D-Limonene adsorbed on cocoa fiber, has been patented (Patent application EP 3097921) and has been registered with the commercial names of Limenorm^®^ and ThangeComplex^®^. It was provided by Targeting Gut Disease Srl (Bologna, Italy). FLS has a total content of D-Limonene of 13.5 ± 0.5 (g/100 g) and total dietary fiber of 54.0 ± 5.7 (g/100 g), composed by 43.1 ± 4.5 (g/100 g) of insoluble part and 10.9 ± 1.2 (g/100 g) of soluble part. For oral administration, FLS was resuspended in 200 µL of glycerol and 200 µL of distilled water.

### 2.2. D-Limonene and FLS Pharmacokinetic Analysis

Adult male Sprague-Dawley rats (200–250 g body weight, Charles River Laboratories, Lecco, Italy) were randomized into two experimental groups. Each group was composed by 4 rats. The fasted rats for 12 h received an oral dose of 200 mg/kg D-limonene (about 50 mg) in two types of formulations: The first group received by gavage D-limonene dissolved in 500 μL of corn oil; the second group received D-limonene adsorbed on 350 mg of cocoa fiber (FLS). At the end of the oral administrations and at fixed time points, blood samples (100 μL) were collected and immediately spiked with 200 μL of acetonitrile and 100 μL of internal standard (100 μM GER-UDCA) dissolved in acetonitrile. After double centrifugation (5 min at 13,000× *g*) of the samples, 10 μL of the supernatant were injected into the HPLC system for D-limonene and GER-UDCA detection. The chromatographic apparatus consisted of a modular system (model LC-10 AD VD pump and model SPD-10A VP variable wavelength UV−Vis detector; Shimadzu, Kyoto, Japan) and an injection valve with a 20 µL sample loop (model 7725; Rheodyne, IDEX, Torrance, CA, USA). Separations were performed at room temperature on a 5-µm Hypersil BDS C-18 column (150 × 4.6 mm i.d.; Alltech Italia Srl, Milan, Italy) equipped with a guard column packed with the same Hypersil material. Data acquisition and processing were performed on a personal computer using CLASS-VP Software, version 7.2.1 (Shimadzu Italia, Milan, Italy). The detector was set at 205 nm; the mobile phase consisted of an isocratic mixture of water and acetonitrile at a ratio 20:80 (*v/v*). The retention times were 4.9 min for D-Limonene and 8.4 min for GER-UDCA [17], used as internal standard for the quantification of D-Limonene in blood samples.

The chromatographic precision, represented by relative standard deviations (RSD), was evaluated by repeated analysis (*n* = 6) of the same sample dissolved in a water and acetonitrile mixture 25:75 (*v/v*) containing D-Limonene at a concentration of 10 µM (1.36 μg/mL). The RSD value was 0.89%. D-Limonene was quantified by the peak area correlated with the predetermined standard curve over the range 0.5–200 µM (0.068–27.2 μg/mL). The calibration curve was linear (*n* = 8, *r* = 0.998, *p* < 0.001).

Recovery experiments of 5 µg/mL D-Limonene from rat whole blood were performed by comparison of the peak areas extracted from blood test samples at 4 °C (*n* = 6) with those obtained by injection of an equivalent concentration of analyte dissolved in a water and acetonitrile mixture 25:75 (*v/v*). The average recovery ± SD was 86.5 ± 3.2%. The concentrations of this compound were therefore referred to as peak area ratio with respect to the internal standard GER-UDCA. The precision of the method, evaluated by replicate analyses (*n* = 6) of rat blood extract containing the internal standard (GER-UDCA) and D-Limonene at a level of 10 μg/mL, was demonstrated by the RSD value of 1.15%. Calibration standards were prepared by spiking the purified blood samples with the internal standard (GER-UDCA) and with known amounts of D-Limonene corresponding to blood concentrations in the range 0.5–100 μM (0.068–13.62 μg/mL) at 4 °C. These solutions were analyzed by HPLC and the calibration curve of peak area ratios versus concentrations was linear (*n* = 8, *r* = 0.994, *p* < 0.001). The in vivo experiments were performed in accordance with the European Communities Council Directive of September 2010 (2010/63/EU). Any effort was done to reduce the number of the animals and their suffering.

### 2.3. Animal Treatment

Forty male C57BL/6 mice, 8 weeks old, were purchased from Charles River Laboratories (Lecco, Italy). Animals were housed in collective cages with a controlled environment containing two mice each, at 22 ± 2 °C and 50% humidity, under a 12 h light/dark cycle. Mice were allowed to acclimate to these conditions for 7 days before inclusion in experiments and had free access to food and water throughout the study. Mice were randomized into four experimental groups. Each group was composed by 8 mice. The first group (Control) received a standard diet (4RF25, Mucedola, Milan, Italy); the second group (HFD) received a HFD (Mucedola, Milan, Italy), composed as in Table 1, and a solution of 200 µL of glycerol and 200 µL of distilled water by oral gavage for 84 days; the third group (HFD + FLS 30) received a HFD and FLS by oral gavage at the dose of 30 mg/die for 84 days; and the fourth group (HFD + FLS 60) received a HFD and FLS by oral gavage at the dose of 60 mg/die for 84 days. Mice were allowed to access their diets ad libitum. All treatments were given to fasting mice at the same hour in the morning starting on day 1 through day 84 of the experiment. On day 84, mice were sacrificed by cervical dislocation after isoflurane anesthetization, a complete necroscopy was performed by a veterinarian to evaluate any macroscopic alteration in the organs. Moreover, samples of colon, stomach, liver, kidney and brain were collected and fixed in formalin for histological evaluation. Animal housing and experimental procedures were performed by the Zooprophylactic Institute of Teramo, Italy, in accordance with European and Italian guidelines. The experimental protocol was approved by the Italian Ministry for Research (Aut. n. 355/2019-PR).

### 2.4. Blood Lipidic Profile and Glucose Measurement

In euthanized mice, blood was drawn by cardiac puncture and immediately transferred into chilled tubes containing a final concentration of 1 mg/mL of ethylenediaminetetraacetic acid (EDTA). Then, the samples were centrifuged at 3000 rpm for 10 min, and the obtained plasma was stored at −80 °C until analysis. The levels of triglycerides, total cholesterol and HDL-cholesterol in the plasma were evaluated by using the automatic biochemical analyzer (ILab 600, Instrumentation Laboratory, Milan, Italy). Glycemia was measured in whole blood by using a glucometer and test strips ACCU-CHEK^®^ Aviva (Roche, Basilea, Switzerland). One drop of blood collected from tail vein was added on a strip inserted in the device; all measurements were performed in fasting mice.

### 2.5. Determination of Plasma Chemokine Levels and Hormonal Profile

Blood for cytokine analysis was collected from the sub-mandibular plexus, and circulating cytokines were quantified by using a customized detection panel purchased from Bio-techne (Minneapolis, MN, USA). The inflammatory cytokines evaluated were IL-1β, IL-6, TNFα, IL-10, IL-17a, IFNγ and MCP-1. The hormonal profile consisted of resistin, adiponectin, IGF1 and leptin. The assays were performed in 96-well filter plates by multiplexed Luminex^®^ (Austin, TX, USA) immunoassay following the manufacturer’s instructions, as previously described [18]. Microsphere magnetic beads coated with monoclonal antibodies against the different target analytes were added to the wells. After 30-min incubation, the wells were washed, and biotinylated secondary antibodies were added. After incubation for 30 min, beads were washed and then incubated for 10 min with streptavidin-PE conjugated to the fluorescent protein, phycoerythrin (streptavidin/phycoerythrin). After washing, the beads (a minimum of 100 per analyte) were analyzed in the BioPlex 200 instrument (BioRad^®^, Hercules, CA, USA). Sample concentrations were estimated from the standard curve using a fifth-order polynomial equation and expressed as pg/mL after adjusting for the dilution factor (Bio-Plex Manager software 5.0). Samples below the detection limit of the assay were recorded as zero, while samples above the upper limit of quantification of the standard curves were assigned the highest value of the curve. The intra-assay CV averaged 15%.

### 2.6. Histological Evaluation of Hepatic Steatosis and Lipid Profile

For histological evaluation, liver samples were collected on day 84, fixed in formalin and embedded in paraffin. Four micrometer sections were stained with hematoxylin-eosin and observed for histological assessment of steatosis by a pathologist in a blinded manner.

### 2.7. Hepatic Tissue Lipid Profile (Lipidomic Analysis)

6cis-16:1 methyl ester, 7cis-16:1 methyl ester, 8cis-18:1 methyl ester and 5cis,8cis methyl ester were purchased from Lipidox (Lidingö, Sweden); cis and trans fatty acid methyl esters (FAME), dimethyl disulfide, iodine, cholesterol, sphingomyelin and formic acid were purchased from Merck-Sigma-Aldrich (Milan, Italy) and used without further purification. Chloroform, methanol, isopropanol, diethyl ether and n-hexane (HPLC grade) were purchased from Baker (Phillipsburg, NJ, USA.HPLC grade) and used without further purification. Silica gel analytical thin-layer chromatography (TLC) was performed on Merck silica gel 60 plates, 0.25 mm thickness, and spots were detected by spraying the plate with cerium ammonium sulfate/ammonium molybdate reagent.

For lipidomic analysis, samples from liver were frozen in liquid nitrogen after collection. The evaluation of fatty acids in lipid extracts from hepatic tissue was performed as previously described in details [19,20]. FAME were quantified based on standard calibration curves. Three samples of hepatic tissue weighing 300 mg from each group, each corresponding to a different mouse, were homogenized. Lipids were extracted from the homogenates using an organic solvent according to the Folch method [21] and subsequently trans esterified to obtain FAME that were characterized by GC and GC/MS.

(FAME) were analyzed by GC (Agilent 6850, Agilent, Santa Clara, CA, USA) in splitless mode, equipped with a 60 m × 0.25 mm × 0.25 µm (50%-cyanopropyl)-methylpolysiloxane column (DB23, Agilent, Santa Clara, CA, USA), and a flame ionization detector, with the following oven program: Temperature started from 165 °C, held for 3 min, followed by an increase of 1 °C/min up to 195 °C, held for 40 min, followed by a second increase of 10 °C/min up to 240 °C, and held for 10 min. A constant pressure mode (29 psi) was chosen with helium as the carrier gas. Methyl esters were identified by comparison with the retention times of authentic samples; The FAME are expressed in quantitative relative percentages (mean ± SD) calculated on the basis of calibration curves of standard references.

Dimethyl disulfide adducts of FAME were analyzed by GC-MS (Thermo Scientific Trace 1300) equipped with a 15 m × 0.25 mm × 0.25 µm 5% phenyl methyl polysiloxane column (TraceGOLD™ -SQC, Thermo Scientific, Waltham, MA, USA) with helium as the carrier gas, coupled to a mass-selective detector (Thermo Scientific ISQ, Waltham, MA, USA) with the following oven program: Temperature started at 80 °C, maintained for 2 min, increased at a rate of 15 °C/min up to 140 °C, increased at a rate of 5 °C/min up to 280 °C and held for 10 min.

In order to understand the position of the 16:1 double bond in the fatty acid 16:1 and thus distinguish the two positional isomers delta 6 and delta 7, 50 μL of dimethyl disulfide (DMDS excess) were added to the FAME mixture and two drops of a 6% solution of iodine in diethyl ether were added in order to obtain the corresponding dimethyl sulfides. After one night at room temperature, a 5% aqueous solution of sodium thiosulfate (3 × 1 mL) was added. The organic component was washed with brine (2 × 1 mL), dried on anhydrous Na_2_SO_4_ and evaporated to dryness.

### 2.8. Characterization of the Fecal Microbiota

Fecal samples were collected from standard cages where animals were housed during the experiments on days 1, 14, 28, 42, 56, 70 and 84 and stored at −80 °C until DNA extraction. Fecal samples from single experimental group were pooled, and Nucleic acids were extracted from 250  mg of sample using PowerSoil^®^ DNA Isolation Kit (MoBio Laboratories, Inc., Carisbad, CA, USA) according to the manufacturer’s recommendations. DNA quality was checked using a Nanodrop 100™ (NanoDrop Technologies, Wilmington, DE, USA). The V3-V4 hypervariable regions of the 16S rRNA gene were amplified using the universal primers 341F and 785R, and then sequenced on an Illumina MiSeq (Illumina, San Diego, CA, USA) platform as per the manufacturer’s instructions, at Wellmicro Srl (Bologna, Italy). All sequence data were processed using a pipeline combining PANDASeq [22] and QIIME 2 [23]. Briefly, quality-filtered reads were binned into Amplicon Sequence Variants (ASVs) using DADA2 [24]. Singletons and chimeras were removed. Taxonomy assignment was performed using the VSEARCH algorithm [25] and the Greengenes database (May 2013 release). Alpha diversity was computed using the inverse Simpson index. Beta diversity was estimated by calculating Bray–Curtis distances between genus-level microbial profiles, which were then used as input for Principal Coordinates Analysis (PCoA).

### 2.9. Statistical Analysis

Continuous variables are expressed as mean ± SD. Normality of distribution was verified with the D’Agostino-Pearson and Shapiro–Wilk tests and the homogeneity of variances (homoscedasticity) with the F-test. Statistical differences between groups were determined by one-way analysis of variance (ANOVA). Tukey’s method was applied as a post hoc test. GraphPad Prism 6 (GraphPad Software Inc., San Diego, CA, USA) was used for all analyses. Differences were considered significant at *p* < 0.05. PCoA and the Adonis test (permutation test with pseudo-F ratio) were carried out using the vegan package of R [26]. For taxonomic comparisons, only taxa whose relative abundance increased or decreased in a given group by at least 2 times were considered.

## 3. Results

### 3.1. D-Limonene and FLS Pharmacokinetic Analysis

Oral administration of D-Limonene (200 mg/kg) in the free form (i.e., dissolved in corn oil) leads to an increase in its blood concentrations, detectable by HPLC measurements, showing a maximum peak (C_max_) of 2.31 ± 0.44 μg/mL at 30 min, as reported in Figure 1. The oral administration of an identical dose of Limonene adsorbed on cocoa fiber (FLS) does not lead to any HPLC detectable blood peak, demonstrating that the D-Limonene contained within the FLS completely loses its bioavailability and do not reach the bloodstream.

### 3.2. Effect of FLS on Weight Gain

Mouse weight was recorded every two weeks (Figure 2). After 14 days of treatment, the HFD + FLS 60 group showed significantly less weight gain than the HFD group, with a difference of approximately 4% (24.73 ± 1.71 g vs. 26.43 ± 1.30; *p* = 0.022). This difference remained significant until the end of the experiment (day 84, 35.24 ± 4.56 vs. 40.79 ± 3.28; *p* = 0.0069). Weight gain in the HFD + FLS 30 group was not significantly different from the HFD group. Overall, the effect of FLS on weight gain was evident only at the daily dose of 60 mg/kg.

### 3.3. Effect of FLS on Food Intake

Food consumption was recorded every two weeks as well (Figure 3). HFD mice overall ate less than those fed with a standard diet. This was expected as the HFD diet is much more caloric and higher in fat, capable of generating a stronger sense of satiety than the standard diet. FLS at both doses tested, significantly reduced food intake compared to HFD mice. This effect on food consumption was significant from 14 days of treatment over the duration of the experiment for the HFD + FLS 60 group, while mice treated with FLS at 30 mg/kg die (HFD + FLS 30 group) showed significant lower food intake values only on days 28, 56 and 70. Overall, FLS was effective in reducing food intake in mice, particularly at the higher dosage.

### 3.4. Effect of FLS on Lipid Profile

Mouse lipid profile was evaluated only at the end of the experiment due to the large amount of plasma required for this type of analysis. After 84 days of FLS treatment, all HFD-fed groups showed significantly higher plasma total cholesterol and HDL-cholesterol values than the control lean group, and no significant effects on these plasma lipids were observed in FLS treated groups (Figure 4). FLS treatment caused a statistically significant reduction in triglyceridemia only in the HFD + FLS 60 group (*p* = 0.0268), with values comparable and non-statistically different from those recorded in the control group.

### 3.5. Effect of FLS on Fasting Glycemia

Blood glycemia was recorded every two weeks (Figure 5). The HFD diet led to an increase in fasting glycemia over time, probably due to the establishment of insulin resistance in these animals. After 14 days of treatment, there was a significant decrease in fasting glucose in both FLS-treated groups compared to the HFD one. This significant decrease in blood glucose was maintained for all experimental points (with the only exception of day 56) in the HFD + FLS 60 group, while in the HFD + FLS 30 group it returned significant only at the end of the experiment, on days 70 and 84.

### 3.6. Effect of FLS on Circulating Cytokines, Adipokines and Hormones

We analyzed the effects of FLS on the hormonal profile of mice, consisting of blood resistin, adiponectin and leptin. While the HFD diet significantly increased leptin and resistin plasma levels, FLS at either dosages did not significantly change their circulating levels (See Appendix A). Since HFD-fed mice show systemic inflammation, [27] we also evaluated the circulating levels of TNFα and IL-6. The HFD diet increased systemic inflammation, as expected, but FLS, at both doses, was not able to significantly reduce this mild inflammation (Appendix A).

### 3.7. Effect of FLS on Hepatic Steatosis of HFD Mice

After 84 days of HFD, severe liver steatosis was visible in fatty mice, with macro lipid droplets mainly represented in hepatocytes (Figure 6A). Less severe steatosis was observed in the HFD + FLS 30 group, with both micro and macro droplets still represented (Figure 6B). In the HFD + FLS 60 group, the hepatic tissue showed a drastic decrease in lipid accumulation (Figure 6C) with a histological structure very similar to that of the control group (Figure 6D).

### 3.8. Effect of FLS on Hepatic Lipids of HFD Mice

The lipid liver profile of all experimental groups is reported in Table 2. HFD induced an increase in monounsaturated fatty acids (MUFAs) and a decrease of saturated (SFAs) and polyunsaturated (PUFAs) fatty acids, both ω-6 and ω-3. The SFAs/MUFAs ratio strongly decreased while the ω-6/ω-3 ratio increases significantly. The latter is an alteration that is typically also found in humans with MS [28]. In the HFD group, there was also a significant increase in trans fatty acids, which are considered to be good indicators of liver stress [29]. The administration of the supplement at low doses was able to significantly decrease the total saturated fats, which accumulated in the liver due to the diet, but which were converted to monounsaturated as a metabolic response of the liver that actively works against an excess of saturated fatty acids. As for saturated fats, the effect differed between palmitic acid (16:0) and stearic acid (18:0): Palmitic acid decreased in mice treated with FLS at the lower dose, both compared to the average value observed in Controls and with respect to the value observed in the HFD group, while for stearic acid there was a significant rebalancing with respect to the HFD group, which tended to bring it back to values closer to those observed in the Control group. With regard to monounsaturated fatty acids, there were slight effects, with a decrease in oleic acid, but not significant, and a significant increase in erucic acid 20:1, which suggests a possible positive effect of the supplement on the elongase enzyme. As for PUFAs, FLS at low doses determined increased values of ω-6 (both DGLA and arachidonic acid) and ω-3 (in particular DHA), which were strongly decreased in hepatocytes in mice that underwent HFD. ARA and DHA were significantly higher in treated mice than in untreated mice, with ARA reaching a level very similar to that found in Controls. In addition, the values of trans isomers, which are a marker of liver stress [30], in treated mice tended to be similar to those of Controls. Regarding the high dosage of the food supplement, it was able to balance the level of saturated fats more than the low dosage, with the main effect on stearic acid, which was similar to the level of Control group. The always low values of palmitic acid compared to the Controls ensured that the SFAs/MUFAs ratio was always kept below 1. FLS decreased the amount of saturated (oleic acid) levels in a dose dependent manner, with significance reached only at 60 mg. Moreover, at both doses FLS significantly increased ω-6 PUFAs which were decreased by HFD diet. A similar trend was observed for DHA. Total trans at the high dose of the food supplement group were however similar to those present in the HFD group, indicating a possible “paradoxical” effect that chronic use of this dosage may have had on mice.

### 3.9. Effect of FLS on Fecal Microbiota Composition in HFD Mice

The 16S rRNA gene sequencing of fecal samples (1 per time point for each group) generated 1,351,843 reads (mean ± SD, 48280 ± 9163), which were clustered into 1771 ASVs. Principal coordinate analysis (PCoA) of inter-individual variation, based on Bray–Curtis distances between genus-level profiles, showed significant separation between Control and HFD group samples, regardless of FLS administration (*p* = 0.001, permutation test with pseudo-F ratio) (Figure 7A). In line with the available literature [31,32], this confirms a major effect of HFD on the gut microbiota structure. As expected, albeit in the absence of statistical significance, the Control group was also characterized by greater alpha diversity (Figure 7B), recognized as a hallmark of health and whose decrease is observed in numerous pathological contexts, including metabolic disorders [33]. Interestingly, the HFD + FLS 60 group showed overall higher biodiversity than the HFD + FLS 30 group and also the HFD group, particularly after 14 and 28 days of treatment. Regarding beta diversity, as anticipated above, there were no evident differences, but the HFD + FLS 30 and HFD + FLS 60 samples tended to separate over time with respect to the HFD counterpart, starting after 14 days of treatment, which might suggest an effect, albeit limited, of FLS.

Taxonomically, all samples were dominated by Firmicutes (mean relative abundance in the whole cohort, 80.5%), with minor proportions of Actinobacteria (8.4%), Verrucomicrobia (5.6%) and Bacteroidetes (3.6%) (Figure 8A). As expected for mice [34], the most abundant family was *Lactobacillaceae* (36.5%), followed by unclassified Clostridiales members (8.4%), *Coriobacteriaceae* (8.4%), *Erysipelotrichaceae* (7.9%), *Staphylococcaceae* (7.5%), *Verrucomicrobiaceae* (5.6%) and *Lachnospiraceae* (5.5%) (Figure 8B). Consistently, the most represented genus in all groups was *Lactobacillus* (36.5%), along with *Adlercreutzia* (6.8%), *Staphylococcus* (6.0%), *Allobaculum* (5.9%) and *Akkermansia* (5.6%) (Figure 8C).

As mentioned above and previously [35,36], HFD induced several alterations in the microbiota composition, especially a marked increase in families known to be associated with inflammation and obesity (i.e., *Peptostreptococcaceae* (mean relative abundance in the Control vs. HFD group, 0% vs. 5.8%), *Erysipelotrichaceae* (4.3% vs. 12.1%) and *Desulfovibrionaceae* (0.04% vs. 0.3%)) (Figure 8B). Within them, it is worth noting that the main discriminant genera were *Allobaculum* and *Coprobacillus* for *Erysipelotrichaceae*, and *Bilophila* for *Desulfovibrionaceae* (Figure 8C). On the other hand, compared to the Control group, the HFD one was depleted of commensal, generally health-associated taxa [37], such as Bacteroidales (*Prevotellaceae*, 2.0% vs. 0%; S24–7, 5.0% vs. 1.4%), Bacillales (*Bacillaceae*, 0.9% vs. 0.1%; *Planococcaceae*, 6.8% vs. 0%) and Clostridiales (unclassified taxa, 9.4% vs. 5.4%; *Clostridiaceae*, 0.3% vs. 0.05%) members, as well as *F16* (1.2% vs. 0.2%).

As for the effect of FLS, it must be said that some dysbiotic signatures were already reversed at low dosage, while others only at the higher dosage (Figure 8). In particular, we observed a decrease in *Peptostreptococcaceae* in both the HFD + FLS 30 and HFD + FLS 60 groups (mean relative abundance in HFD + FLS 30 vs. HFD + FLS 60 vs. HFD group, 0.01% vs. 2.5% vs. 5.8%), and in *Desulfovibrionaceae* and *Erysipelotrichaceae* genera in the HFD + FLS 60 group only (HFD + FLS 60 vs. HFD group, 0.08% vs. 0.2% for unclassified *Desulfovibrionaceae*; 1.5% vs. 8.3% for *Allobaculum*), especially after 28 days of treatment. It should be noted that the mean relative abundance of the *Erysipelotrichaceae* family in the HFD + FLS 60 group (3.6%) approximated that of the control group (4.3%). On the other hand, both HFD + FLS 30 and HFD + FLS 60 groups shared an increased representation of *Bacillaceae* (HFD + FLS 30 vs. HFD + FLS 60 vs. HFD, 0.3% vs. 0.5% vs. 0.1%), *Planococcaceae* (1.6% vs. 2.8% vs. 0%) and *Clostridiaceae* (2.9% vs. 3.0% vs. 0.05%). For the latter, the increase was observed starting from 28 days of treatment, while for the other families, the increase (mainly attributable to *Bacillus* and *Sporosarcina*) was particularly evident in the HFD + FLS 60 group towards the end of treatment (i.e., after 56 days of treatment).

## 4. Discussion

D-Limonene and orange EO, rich in this monoterpene, have been used as substances capable of counteracting obesity with multitarget positive effects in rodent models. Orange EO microencapsulated into beta-cyclodextrin has been used in a model of obesity obtained by feeding rats with a HFD [11]. This study demonstrated an anti-obesity effect of orange EO by significantly decreasing body weight gain, alleviating liver pathological alteration and ameliorating HFD-related biochemical parameters, such as the total cholesterol blood level. The molecular mechanisms involved in these positive effects appeared to be decreased expression of peroxisome proliferator-activated receptor (PPAR)-γ in the liver, upregulation of uncoupling protein 2 and increased expression of hormone-sensitive lipase and carnitine palmitoyltransferase I in the adipose tissue. Moreover, reduced insulin levels were observed in orange EO-treated mice. The doses at which these effects were recorded were 19 mg of EO daily, corresponding to 70–80 mg/kg body weight in rats. Reduced insulin resistance was previously achieved in a HFD rat model, by using a D-Limonene-enriched diet [12]. In this model, the presence of 2% D-Limonene in the diet reversed the HFD-induced pathological alteration in the liver and pancreas. Orange EO and D-Limonene in particular were found to be capable of modulating the intestinal microbiota of mice, by increasing the relative abundance of *Lactobacillus* and changing the bacterial compositional structure in the cecum and colon [10]. This bacterial shift was also linked to a decrease in short-chain fatty acids synthesis by the mouse microbiota, which was upregulated in obese mice. D-Limonene administered at 600 mg/kg (dissolved in 0.1% DMSO) to HFD mice was capable of reducing metabolic disorders linked to obesity [38]. In particular, D-Limonene-treated HFD mice exhibited lowered serum triglyceride and fasting blood glucose levels, and decreased liver lipid accumulation. At the molecular level, D-Limonene treatment activated PPAR-α, and inhibited liver X receptor (LXR)-β signaling.

Thus, it is not surprising that D-Limonene has been proposed as an ingredient for food supplements aimed at benefitting patients with obesity and metabolic syndrome. It could really be an active dietary supplement ingredient to prevent and ameliorate dyslipidemia, hyperglycemia and metabolic disorders. Nevertheless, toxic D-Limonene dosages are not much different from those in which D-Limonene exerts its protective effects against the damages caused by HFD diets [39]. Probably for this reason, no clinical trials have been carried out on humans with orange EO or D-Limonene at daily dosages that, using an allometric scaling, would be equivalent to 10–20 mg/kg body weight, close to the maximum dose level value of 25 mg/kg suggested by EFSA for all animal species in the oral route. Our FLS formulation significantly decreased D-Limonene absorption in the small intestine, concentrating its action inside the gut and on its microbiota, thus reducing possible systemic adverse effects, such as liver toxicity. Because of the low bioavailability of FLS, we did not observe some of the systemic effects that D-Limonene exerted in other studies on obese HFD mice, such as lowering cholesterolemia or reducing systemic inflammation. On the other hand, we probably maximized the effect of this monoterpene on the gut microbiota. In fact, we observed some interesting trends already after two to four weeks of treatment. FLS (especially at the higher dosage) tended to reverse the main alterations induced by HFD, positively modulating the compositional structure of the microbiota. Indeed, we found decreased proportions of potentially pro-inflammatory taxa, such as *Desulfovibrionaceae* (known sulphate-reducing pathobionts), *Peptostreptococcaceae* and *Erysipelotrichaceae*, which are typically increased in metabolic inflammatory disorders [40]. On the other hand, we found increased amounts of *Bacillaceae*, *Planococcaceae* and *Clostridiaceae*, which tended to approximate the values of control mice. These results only partially overlap with those of Wang and collaborators [10], who had mainly observed an increase in the relative abundance of *Lactobacillus* after administration of orange EO and D-Limonene, but it should be noted that in our study FLS was given to mice fed a HFD diet, which makes the data not directly comparable. It should also be remembered that FLS is also composed of fibers and therefore some of the effects we have observed could be linked to its prebiotic activity. The prebiotic effects of FLS and a standard prebiotic such as inulin have recently been compared [16]. This comparison clearly showed that FLS activities on the human microbiota were largely driven by its D-Limonene content. Therefore, it is possible to consider that the activities of FLS largely depend on D-Limonene.

Despite the permanence of FLS inside the intestine, there is no doubt that it produced systemic effects, in particular on the body weight of the animals, on their food intake, on fasting glycemia and triglyceridemia, but above all on liver steatosis. An evident dose-dependent reduction of steatosis was obtained for both FLS doses, associated with a strong improvement in the hepatic lipidomic profile, which instead did not appear to be dose-dependent, being present at the lowest dose. The known properties of Limonene to act as antioxidant and protect PUFA are used in food manufacturing [41]. The antioxidant and antimicrobial activities of these components are attracting interest for application in agri-food [42]. In our experiments, the increase of PUFA detected in liver tissues could be due to an increased in vivo protection by Limonene, as active substance toward the composition of the lipid pool.

The strengths of this study are having formulated and preclinically tested a food supplement containing high dosages of D-Limonene, but with reduced intestinal absorption in order to minimize its systemic and hepatic toxic effects. One of the limitations of this study is that it did not find correlations between the reduction of mouse body weight in FLS-treated groups and modifications in the secretion of adipokines, such as resistin, adiponectin, IGF1 and leptin, whose levels are altered by the HFD diet. Another limitation is that we have focused on the compositional alterations of the microbial ecology and not on functional implications, in terms of changes in the pool of bioactive molecules produced by the microbiota. However, interesting insights in this direction have recently been provided through an in vitro model using human microbiota samples [16]. Finally, it should be noted that we did not find a significant reduction in the inflammatory profiles even if the microbiota data suggest a reduction in families typically associated with increased inflammation. These data could appear conflicting; however, it should be noted that in the HFD group we observed only mild inflammation while we recorded a strong steatosis, reversed by FLS. Thus, probably, in this model, the liver metabolism and the microbiota shift impact on metabolic parameters much more than the systemic inflammation.

## 5. Conclusions

FLS is a food supplement, based on D-Limonene, extremely effective in countering some of the main negative effects of the HDF diet, such as weight gain, hyperglycemia and hyperglyceridemia. These therapeutic effects, which are not new to D-Limonene, have been obtained with a very low intestinal absorption formula, which allows to move from a preclinical to a clinical human study, without incurring possible concerns regarding the systemic toxicity of this monoterpene. FLS is, therefore, suitable for bringing D-Limonene in a study on humans, at daily doses even higher than the ADI.

## Figures and Tables

**Figure 1 pharmaceutics-13-01753-f001:**
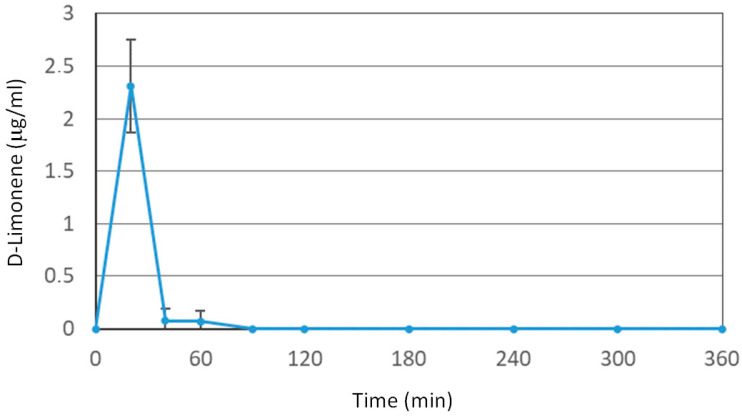
Blood D-Limonene concentrations (μg/mL) within 360 min after oral administration of a 50 mg dose (i.e., 200 mg/kg) to rats. Data are expressed as the mean ± SE of four independent experiments. The oral formulation consisted of D-Limonene dissolved in corn oil.

**Figure 2 pharmaceutics-13-01753-f002:**
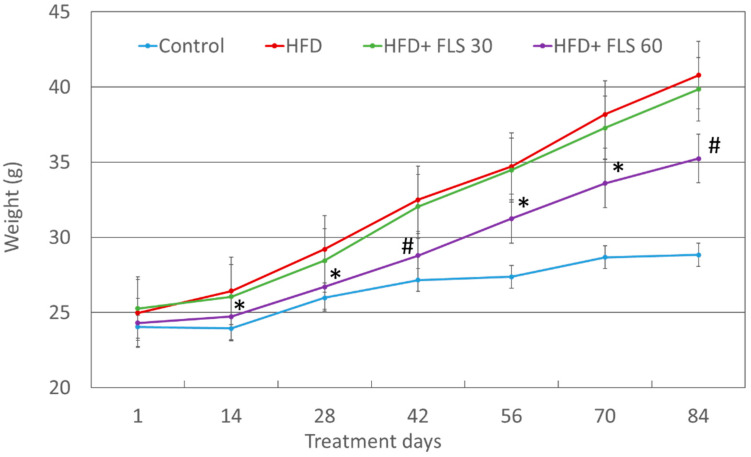
Average weight of the experimental groups of mice throughout the whole duration of the experiment. Error bars represent the standard error of the mean. *, *p* < 0.05; #, *p* < 0.01 compared to the HFD group.

**Figure 3 pharmaceutics-13-01753-f003:**
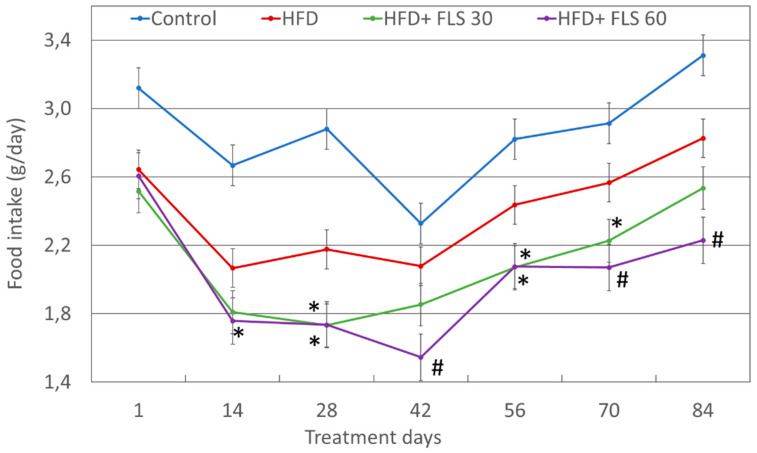
Average per capita food intake trend throughout the whole duration of the experiment. Error bars represent the standard error of the mean. Mice treated with FLS at both dosages showed a reduction in food intake, compared with the HFD group. *, *p* < 0.05; #, *p* < 0.01 compared to the HFD group.

**Figure 4 pharmaceutics-13-01753-f004:**
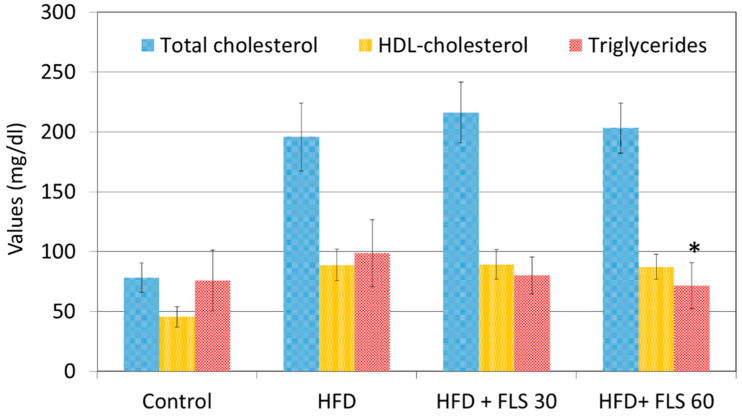
Average total cholesterol, HDL-cholesterol and triglycerides measured in the blood of the mice at the end of the experiment, after 84 days of FLS treatment. Error bars represent the standard error of the mean. FLS at 60 mg/kg significantly reduced the blood triglyceride concentrations to values not statistically different from those measured in the Control group. *, *p* < 0.05 compared to the HFD group.

**Figure 5 pharmaceutics-13-01753-f005:**
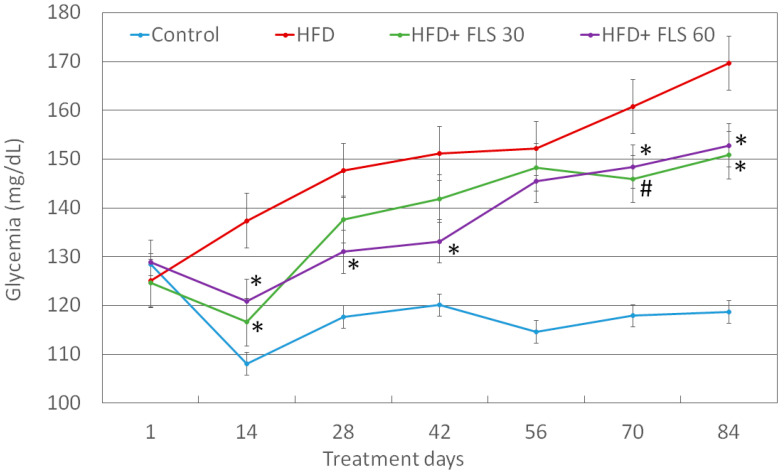
Average fasting glucose measured in the blood of the mice throughout the whole duration of the experiment. Error bars represent the standard error of the mean. *, *p* < 0.05; #, *p* < 0.01 compared to the HFD group.

**Figure 6 pharmaceutics-13-01753-f006:**
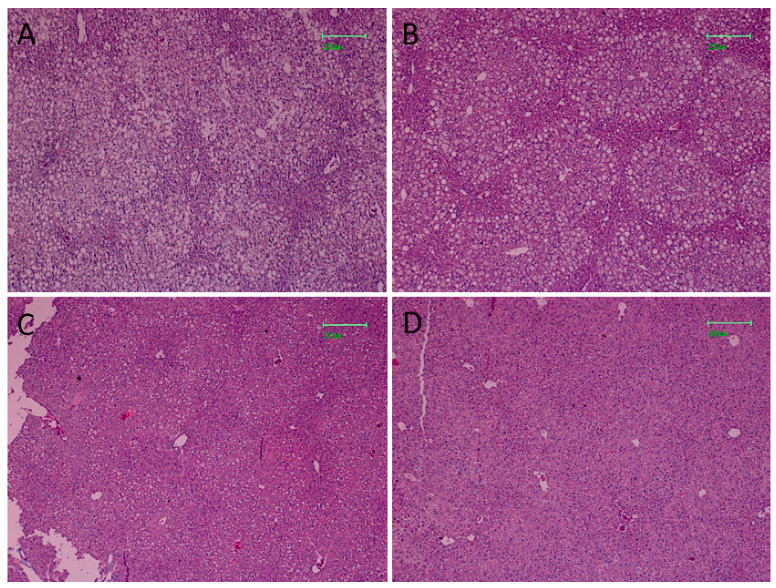
Hepatic histological morphology observed in the HFD group (**A**), HFD + FLS 30 group (**B**), HFD + FLS 60 group (**C**) and Control group (**D**). Magnification: 10×, bars = 200 μM.

**Figure 7 pharmaceutics-13-01753-f007:**
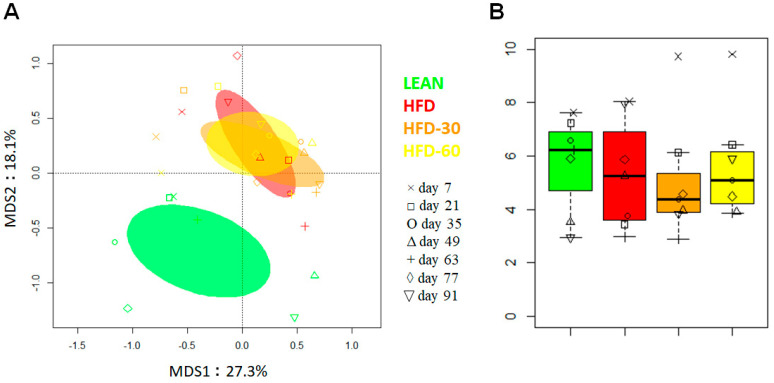
Diversity of the gut microbiota in a mouse model of HFD-induced obesity before and after administration of FLS at different dosages. (**A**) PCoA plot of inter-sample diversity, based on Bray–Curtis distances between the genus-level profiles. Significant separation between the Control group and all other groups was found (*p* = 0.001, permutation test with pseudo-F ratio). Samples are identified with different colors and symbols according to group and time point. Ellipses include 95% confidence area based on the standard error of the weighted average of sample coordinates. (**B**), Boxplots showing the distribution of alpha diversity, according to the inverse Simpson index, in the Control vs. HFD vs. HFD + FLS 30 vs. HFD + FLS 60 groups. Symbols indicate the different time points as in (**A**).

**Figure 8 pharmaceutics-13-01753-f008:**
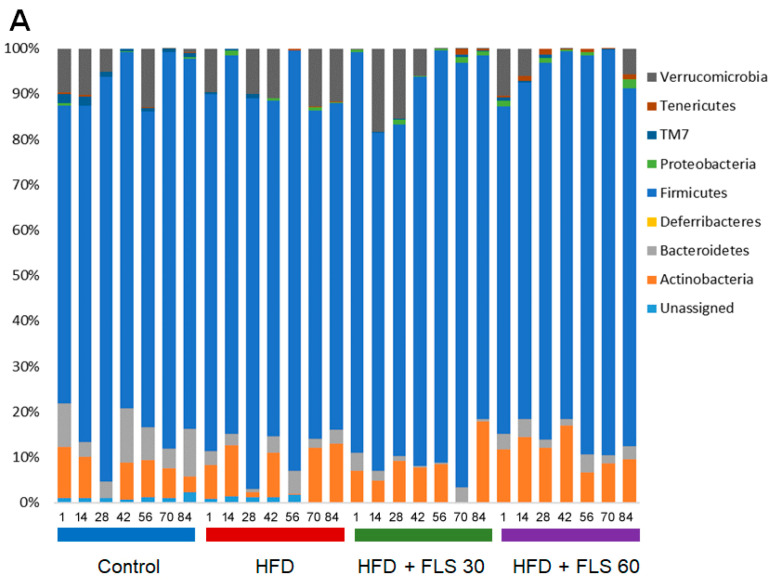
Impact of FLS on the composition of the gut microbiota of HFD-fed mice. Bar plots showing the relative abundance of major phyla (**A**), families (**B**) and genera (**C**) in the gut microbiota of mice receiving a standard diet (blue) or a HFD in the absence (red) or in the presence of FLS at 30 mg/die (green) or 60 mg/die (purple). For each group, the profiles are shown in chronological order (i.e., from day 1 to day 84 of treatment). Only taxa with mean relative abundance ≥0.1 in all mice are shown.

**Table 1 pharmaceutics-13-01753-t001:** Energy density and macronutrients composition of standard diet (STD) and high fat diet (HFD).

Components	STD	HFD
Total Energy, Kcal/g	3.5	6
Protein, %	20	20
Carbohydrate, %	70	20
Fat, %	10	60

**Table 2 pharmaceutics-13-01753-t002:** Analysis of the total lipids present in the liver tissue of Control mice or mice subjected to HFD diet, treated or not treated with FLS. The type of lipids present in the livers of the animals was determined after extraction of the total lipids and derivation into methyl esters. The statistical analysis carried out using ANOVA showed significant differences, indicated in the table with the symbols specified below. HFD vs. Control: ^1^ Relative amount * *p* ≤ 0.047; ** *p* ≤ 0.0089; *** *p* ≤ 0.0005. HFD + FLS 30 vs. HDF: # *p* ≤ 0.025; ## *p* ≤ 0.005; ### *p* ≤ 0.0007. HFD + FLS 60 vs. HFD: § *p* ≤ 0.046; §§ *p* ≤ 0.009. HFD + FLS 60 vs. HFD + FLS 30: £ *p* ≤ 0.0255.

Fatty Acids ^1^	Control	HFD	HFD + FLS 30	HFD + FLS 60
14:0	0.37 ± 0.05	0.47 ± 0.07	0.45 ± 0.03	0.49 ± 0.13
16:0	24.33 ± 0.90	24.25 ± 0.59	20.46 ± 0.35 ###	22.07 ± 0.72 §, £
16:1 6t	0.02 ± 0.00	0.03 ± 0.01	0.02 ± 0.01	0.02 ± 0.01
16:1 9t	nd	nd	nd	nd
16:1 (6 + 7 c)	0.50 ± 0.13	1.73 ± 0.23 **	1.63 ± 0.21	1.31 ± 0.56
16:1 9 c	3.76 ± 0.43	2.96 ± 0.36	2.35 ± 0.30	2.20 ± 0.15 §
18:0	8.61 ± 0.14	4.13 ± 0.40 ***	6.90 ± 1.31 #	8.10 ± 2.31 §
18:1 9 t	0.02 ± 0.01	0.04 ± 0.02	0.03 ± 0.01	0.04 ± 0.02
18:18c	0.08 ± 0.04	0.13 ± 0.08	0.09 ± 0.01	0.09 ± 0.02
18:1 9 c	18.50 ± 2.07	41.55 ± 0.81 ***	36.18 ± 3.36	33.78 ± 3.67 §
18:1 11 c	4.12 ± 0.48	3.57 ± 0.20	3.50 ± 0.37	3.31 ± 0.29
18:2 5c.8 cis	0.05 ± 0.01	0.04 ± 0.00 **	0.05 ± 0.02	0.04 ± 0.02
mt 18:2	0.06 ± 0.02	0.14 ± 0.04 *	0.09 ± 0.01	0.10 ± 0.01
18:2 w6	15.75 ± 0.32	10.41 ± 0.46 ***	10.39 ± 0.19	10.93 ± 1.68
18:3 w 6	0.32 ± 0.01	0.28 ± 0.07	0.18 ± 0.00	0.19 ± 0.08
18:3 w 3	0.45 ± 0.03	0.30 ± 0.07 *	0.27 ± 0.08	0.27 ± 0.07
20:0	0.17 ± 0.03	0.13 ± 0.09	0.18 ± 0.04	0.18 ± 0.04
20:1 9c	0.42 ± 0.10	0.79 ± 0.05 **	0.92 ± 0.03 #	0.75 ± 0.22
20:3 w 6 DGLA	3.20 ± 0.30	1.57 ± 0.14 **	2.65 ± 0.51 #	2.66 ± 0.37 §§
20:4 w 6 ARA	8.49 ± 1.93	4.31 ± 0.40 *	8.66 ± 1.86 #	8.14 ± 1.21 §§
monotrans 20:4	0.05 ± 0.03	0.02 ± 0.01	0.01 ± 0.00	0.06 ± 0.04
20:3 w3 EPA	0.66 ± 0.08	0.17 ± 0.08 **	0.09 ± 0.01	0.08 ± 0.03
22:5 w 3 DPA	0.50 ± 0.05	0.33 ± 0.09 *	0.23 ± 0.08	0.23 ± 0.08
22:6 w 3 DHA	9.49 ± 0.78	2.67 ± 0.19 ***	4.60 ± 0.81 #	3.97 ± 0.76 §
SFA s	33.48 ± 0.91	28.98 ± 0.12 **	27.99 ± 0.99 **	30.84 ± 2.97
MUFAs	27.37 ± 2.32	50.73 ± 1.01 ***	44.67 ± 4.21	41.44 ± 4.86 §
PUFAs	38.92 ± 3.09	20.07 ± 0.98 ***	27.12 ± 3.27 #	26.50 ± 1.84 §§
w6	27.77 ± 2.45	16.57 ± 0.71 **	21.88 ± 2.54 #	21.91 ± 1.26 §§
w3	11.10 ± 0.66	3.46 ± 0.32 ***	5.19 ± 0.80 #	4.55 ± 0.81
SFAs/MUFAs	1.23 ± 0.08	0.57 ± 0.01 ***	0.63 ± 0.08	0.76 ± 0.16
w6/w3	2.50 ± 0.08	4.80 ± 0.32 ***	4.24 ± 0.29	4.90 ± 0.76
Tot trans FAs	0.15 ± 0.04	0.23 ± 0.02 *	0.16 ± 0.01 ##	0.22 ± 0.03 £

## Data Availability

Sequencing data are accessible at the National Center for Biotechnology Information Sequence Read Archive.

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
