# Peer review of "Effect of a Fiber D-Limonene-Enriched Food Supplement on Intestinal Microbiota and Metabolic Parameters of Mice on a High-Fat Diet"

_pharmaceutics, 2021, doi:10.3390/pharmaceutics13111753_

Round 1
Reviewer 1 Report
- The word "in vivo" in the text must be in italics (example: line 136 and 527)
- In Point 2.3. you say: "Forty-eight-week-old-mice". What is the age of animals? and number of animals
- Explain the reason of use rats for pharmacokinetc study and mice for obesity study
- It was unclear the fecal samples were collected, please explain if you used metabolic cages or other method.
- How explain the physiological effects of D-Limonene adsorbed on dietary fibers if it is not bioavailable?
- Check the title of table 2 "fecal microbiota"
Author Response
1. The word "in vivo" in the text must be in italics (example: line 136 and 527)
The words “in vivo” are in italics in the revised manuscript.
2. In Point 2.3. you say: "Forty-eight-week-old-mice". What is the age of animals? and number of animals
We agree that the phrase was unclear. We have changed with “Forty male C57BL/6 mice, 8 week old,”
3. Explain the reason of use rats for pharmacokinetc study and mice for obesity study
We used Rats instead of mice for ethical reasons. The amount of blood necessary to perform pharmacokinetics would have required the mouse sacrified, while in the rat the suffering is limited to blood sampling. The Italian Ministerial Animal Ethics Committee requires that animals with less neurological development be used experimentally. In the case of experimentation with FLS, the mouse is sufficient. Our Commettee would not have approved a pharmacokinetics protocol of this type on mice.
4. It was unclear the fecal samples were collected, please explain if you used metabolic cages or other method.
Fecal samples were collected from standard cages where animals were housed during the experiments. Fecal samples from single experimental group were pooled before Nucleic acids extraction. These details has been added to the methods section.
5. How explain the physiological effects of D-Limonene adsorbed on dietary fibers if it is not bioavailable?
The purpose of our study was to allow the D-Limonene to reach the colon and act locally on the microbiota. We have previously shown that the absorption of essential oil on a fiber matrix strongly decrease its intestinal absorption (see Pavan et al., Geraniol Pharmacokinetics, Bioavailability and Its Multiple Effects on the Liver Antioxidant and Xenobiotic-Metabolizing Enzymes. Front Pharmacol. 2018 Jan 25;9:18. doi: 10.3389/fphar.2018.00018.). In the same way we absorbed D-Limonene on Cocoa fibers to reduce its intestinal absorption, as demonstrated by data presented in Figure 1.
6. Check the title of table 2 "fecal microbiota"
We apologize for this typo. It was not a title but a a sentence left by mistake”.
Reviewer 2 Report
This manuscript describes the effects of a dietary supplement based on D-Limonene adsorbed on cocoa fiber on lipid metabolism and microbiota in mice fed a high-fat diet. The formulation of the supplement has been made with a low dose of D-Limonene, thus reducing its negative effects. In fact, D-Limonene does not reach the bloodstream, so the observed effects must be attributed to changes in the microbiota.
There is a clear effect of the dietary supplement on hepatic steatosis and on the lipid profile. In addition, this supplement shows benefits on body weight and blood glucose. These results are interesting and encouraging for further studies of D-limonene in this formulation.
Major comments
- The experimental design suggests that the observed effects are due mostly to cocoa fiber, as D-Limonene is not absorbed under the conditions under which it has been administered, as indicated by the authors. It should be noted that cocoa fiber influences intestinal fat absorption and lipid metabolism, with demonstrated effects on the reduction of plasma lipid concentration (some examples: Ramos et al., 2008, J Agric Food Chem 56: 6985- 6993; Sánchez et al., 2010, J Med Food, 13: 621-628; see also the work on cocoa by the group of Pérez-Cano et al.). Therefore, the study should be reconsidered based on this premise. On the other hand, the effects of D-Limonene on lipid metabolism and intestinal microbiota are already known, as discussed by the same authors in the introduction and discussion sections.
- Another element that weakens the study is the approach to the analysis of the microbiota. Fecal samples were sequenced at various times in the study, resulting in many conditions, but in each case only data from a single sample are available. The variability of the results of microbiota studies is remarkable and should be considered. The reviewer appreciates the effort involved in working with so many conditions, but the results are inconclusive. No clear patterns can be observed within each experimental group. On the other hand, emphasis is placed on improving the profile of the microbiota, in the sense of promoting a profile compatible with the decrease in systemic inflammation, but there is no correspondence in the results obtained when proinflammatory cytokines were measured.
Minor comments
- Line 88. There is a mistake: “absorbed” should be “adsorbed”.
- Please improve Figures 2, 3, and 5. The axes are not visible and the symbols at each point are too small.
- The format of Figures 1, 2 3 and 5 should be harmonized.
- Line 355-356. The statement “The administration of the supplement at low doses was able to significantly decrease the total saturated fats” is not deduced from the data in Table 2. Please review the text.
- Line 375-379. It seems that DHA should be replaced by FLS. The sentence is difficult to understand. Please review it.
Author Response
Major comments
1. The experimental design suggests that the observed effects are due mostly to cocoa fiber, as D-Limonene is not absorbed under the conditions under which it has been administered, as indicated by the authors. It should be noted that cocoa fiber influences intestinal fat absorption and lipid metabolism, with demonstrated effects on the reduction of plasma lipid concentration (some examples: Ramos et al., 2008, J Agric Food Chem 56: 6985- 6993; Sánchez et al., 2010, J Med Food, 13: 621-628; see also the work on cocoa by the group of Pérez-Cano et al.). Therefore, the study should be reconsidered based on this premise. On the other hand, the effects of D-Limonene on lipid metabolism and intestinal microbiota are already known, as discussed by the same authors in the introduction and discussion sections.
In the cited studies, soluble cocoa fiber product (SCFP) was incorporated in chow at 20% (Ramos et al., 2008) or at 10% (Sanchez et al., 2010). Also in other studies focusing on microbiota, the cocoa fibers were incorporated in the chow at 10% (Cladlera et al., 2015, Cladlera et al., 2017). In our study, FLS was administered, at the dose of 30 and 60 mg/die, corresponding to 25 and 50 mg of cocoa fiber product. Considering the mean daily amount of consumed food which ranges from 1,5 to 3g/day, the cocoa fiber was administered in percentage ranging from 1,6% to 3% of chow at the higher dose (60 mg of FLS) and 0,8% to 1,6 % for the lower dosage (FLS 30 mg). Moreover, FLS was not incorporated to chow, but it was administered once a day in fasting mice. With this timing, and this dosages, cocoa fiber could have only minimally influenced food intestinal absorption. Despite this, we agree that a contribution of cocoa fiber on the observed results is certainly present, but it is unlikely that the observed differences in weight gain and glycemic profile are mainly dependent to cocoa fiber administration. Moreover, FLS is a patented mixture which should be considered as a single food supplement. FLS has sown to be capable to positively modulate the human intestinal microbiota in vitro (Nissen et al., Foods, 2021, 10(10), 2371; doi: 10.3390/foods10102371, in press). Also in this case, it was not very important to dissect the effect of the single component of this supplement. The same happens in a lot of publications in which the effects of a formulated dietary supplement are analyzed, without investigating the effects of their individual components. Finally, we are now using FLS in a double-blind clinical trial on humans controlled with placebo, again without dissecting the effects of D-Limonene and cocoa fiber. This consideration on FLS has been briefly added in the Methods section.
2. Another element that weakens the study is the approach to the analysis of the microbiota. Fecal samples were sequenced at various times in the study, resulting in many conditions, but in each case only data from a single sample are available.
We agree with this criticism. We are aware that sample collection from single mice would have been the optimal condition to study the microbiota trajectory, but it was logistically impossible since single mice should have been housed alone in single cage for all the duration of the experiments, and guidelines for animal welfare do not allow us to keep animals isolated given the nature of the study and its duration. Another way to collect single mice samples could have been to move single animals in cages for one day and collect fecal samples, but, again, this procedure repeated for all the time points would have caused too much stress to animals as well as compromise the experiment reliability. We chose to study the microbial trajectory in more timepoints and collect all fecal content from cages, and this choice was made in an attempt to obtain longitudinal information on the impact of FLS on the gut microbiota, parallel to what was done for other variables, such as weight gain, food intake and glycemia. Despite the data refer to a single sample, which however represents the pool of fecal samples of the mice belonging to the same experimental group, and the limitation of our sample collection strategy, we think that different microbiota patterns are visible. Moreover, we think that these data provide a good rationale to include microbiota analysis in clinical studies involving FLS treatment. We are perfectly aware that this remains a limitation of our study, but on the other hand we believe it may have helped to reduce interindividual variability. In the revised manuscript, at the end of the Discussion section, we have included these considerations.
3.The variability of the results of microbiota studies is remarkable and should be considered. The reviewer appreciates the effort involved in working with so many conditions, but the results are inconclusive. No clear patterns can be observed within each experimental group. On the other hand, emphasis is placed on improving the profile of the microbiota, in the sense of promoting a profile compatible with the decrease in systemic inflammation, but there is no correspondence in the results obtained when proinflammatory cytokines were measured.
We agree with this criticism. It is well known that dysbiosis, characterized by loss of diversity, is associated to obesity. Several experiments, also involving fecal transplantation, showed that microbiota can trigger metabolic impairments typical of metabolic syndrome and that transplantation from lean donors to obese donors can lead to an improvement of metabolic parameters (see for example de Groot et al., Fecal microbiota transplantation in metabolic syndrome: History, present and future. Gut Microbes. 2017, 8:253-267 and also Jiao et al., Gut microbiome may contribute to insulin resistance and systemic inflammation in obese rodents: a meta-analysis. Physiological Genomics 2018 50:4, 244-254). Our treatment was able to counteract the shift of fecal microbiota typically observed in HFD obese mice, and we expected that one of the mechanisms involved in metabolic parameters improvement could have been linked to the decrease of mice systemic inflammation. Nevertheless, we did not find a significant reduction in the inflammatory profiles even if the microbiota data suggest a reduction in families typically associated with increased inflammation. These data could appear conflicting; however, it should be noted that in the HFD group we observed only mild inflammation while we recorded a strong steatosis reversed by FLS. So probably, in this model, the liver metabolism and the microbiota shift are linked to the improvement of metabolic parameters much more than the systemic inflammation is. In agreement with the Reviewer’s remarks, in the revised version of our manuscript, we stressed these points at the end of the Discussion.
Minor comments
1. Line 88. There is a mistake: “absorbed” should be “adsorbed”.
This has been corrected.
2.Please improve Figures 2, 3, and 5. The axes are not visible and the symbols at each point are too small. The format of Figures 1, 2 3 and 5 should be harmonized.
Figures have been improved and their format has been harmonized, following Reviewer's suggestions.
3. Line 355-356. The statement “The administration of the supplement at low doses was able to significantly decrease the total saturated fats” is not deduced from the data in Table 2. Please review the text.
The text was right, in Table 2 it was missing the significance. We apologize for this mistake.
4. Line 375-379. It seems that DHA should be replaced by FLS. The sentence is difficult to understand. Please review it.
DHA has been replaced with FLS. Moreover, we agree that the phrase was not clear, and we have changed it in “FLS decreased the amount of saturated (oleic acid) levels in a dose dependent manner, with significance reached only at 60 mg. Moreover, at both doses FLS significantly increased ω-6 PUFAs which were decreased by HFD diet. A similar trend was observed for DHA."
Round 2
Reviewer 2 Report
Minor comments
Line 537. There is a misspelling in adiponectin.
Line 544. In vitro should be in italics
Lines 537-540. In the methods section you have indicated that the feces were collected directly from the cages and that the results refer to the entire group. I think there is no need to point it out in the discussion section.
Author Response
Minor comments
1) Line 537. There is a misspelling in adiponectin.
The word has been corrected.
2) Line 544. In vitro should be in italics
Corrected.
3) Lines 537-540. In the methods section you have indicated that the feces were collected directly from the cages and that the results refer to the entire group. I think there is no need to point it out in the discussion section.
We agree with this criticism and the phrase pointing out feces collection has been removed from the discussion section.
All these corrections have been highlighted in green in the revised manuscript.